# The scaling laws of edge vs. bulk interlayer conduction in mesoscale twisted graphitic interfaces

Debopriya Dutta [1,3], Annabelle Oz[2,3], Oded Hod[2] & Elad Koren [1✉]

The unusual electronic properties of edges in graphene-based systems originate from the pseudospinorial character of their electronic wavefunctions associated with their non-trivial topological structure. This is manifested by the appearance of pronounced zero-energy electronic states localized at the material zigzag edges that are expected to have a significant contribution to the interlayer transport in such systems. In this work, we utilize a unique experimental setup and electronic transport calculations to quantitatively distinguish between edge and bulk transport, showing that their relative contribution strongly depends on the angular stacking configuration and interlayer potential. Furthermore, we find that, despite of the strong localization of edge state around the circumference of the contact, edge transport in incommensurate interfaces can dominate up to contact diameters of the order of 2 μm, even in the presence of edge disorder. The intricate interplay between edge and bulk transport contributions revealed in the present study may have profound consequences on practical applications of nanoscale twisted graphene-based electronics.

[1] Faculty of Materials Science and Engineering and the Russell Berrie Nanotechnology Institute, Technion – Israel Institute of Technology, 3200003 Haifa, Israel. [2] Department of Physical Chemistry, School of Chemistry, The Raymond and Beverly Sackler Faculty of Exact Sciences and The Sackler Center for Computational Molecular and Materials Science, Tel Aviv University, Tel Aviv, IL 6997801, Israel. [3] These authors contributed equally: Debopriya Dutta, Annabelle Oz. ✉email: eladk@technion.ac.il

Graphene-based materials have been a subject of interest for more than a decade due to their wealth of superior and exotic physical properties, which stem mainly from their bulk two-dimensional hexagonal $SP_2$ type honeycomb lattice[1–3]. Upon stacking two graphene sheets to create a bilayer, promising features, such as the ability to open bandgaps by applying vertical electric fields[4–6] and the appearance of superconductivity in magic-angle twisted bilayers[7,8], have been experimentally demonstrated for bulk material transport. Further unique physical properties become accessible when considering finite graphitic systems that bare exposed (or chemically passivated) edges. These can range from quantum confinement effects exhibited by narrow armchair graphene nanoribbons[9,10] to the appearance of strongly confined electronic states along zigzag graphene edges corresponding to a sharp zero-energy peak in their density of states[11–16]. Many fascinating phenomena originating from the properties of such edge states in graphene layers have been predicted theoretically, including electric-field-tunable magnetism[12] and valley-dependent transport[17,18], and where further confirmed to withstand strong edge disorder[14,19]. Furthermore, the importance of edge transmission with respect to bulk conductance has been discussed in the context of graphene bilayer-based logic devices[6], where edge leakage has been suggested to limit their applicability[20,21]. Similar concerns hold for topological insulators, where the material bulk presents a bandgap, whereas the surface exhibits high conductivity[22]. Consequently, both from a scientific perspective and from a technological viewpoint, it is of key importance to decipher the interplay between edge and bulk transport properties in graphene-based systems.

Recently, using a carefully designed experimental setup, we have been able to gain precise control over the stacking configuration of a single twisted graphitic interface. This allowed us to study the interplay between the structure of the interface and its transport properties with high angular resolution i.e. ~ 0.1°[23]. Nevertheless, the critical role that the system edges play in dictating the physical properties of the entire interface remains to be revealed.

To address this question, in the present study, we investigate the separate role of edge and bulk transport in twisted bilayer graphene interfaces. To this end, we study the interlayer charge transport across twisted bilayer graphitic interfaces by means of electromechanical manipulation of nano-sized contacts. To distinguish between bulk and edge contributions, we rely on the distinct dependence of the interfacial transport across the graphitic junction on its cross-section area. Since the edge and bulk overlap areas scale differently with respect to relative lateral shifts of the finite interface, careful manipulation of the junction allows us to extract their individual contributions. We find that, despite the naive expectation that bulk conductance should dictate the transport properties of mesoscale interfaces, pronounced edge states can dominate the system's behavior up to a contact diameter as large as 2 μm.

## Results

**Experimental analysis**. To demonstrate this, we constructed graphitic structures from highly oriented pyrolytic graphite (HOPG) based on a recently presented fabrication method[23–27]. Samples featuring cylindrical structures with a typical height of 50 nm and a diameter of 300 nm were fabricated by means of reactive ion etching, using structured Pd–Au metal layers as self-aligned shadow masks (10- and 40-nm-thick Pd and Au layers, respectively). We use atomic force microscopy (AFM) under ambient conditions to shear individual nano-sized graphitic contacts and to measure the lateral shear forces and current modulations during their mechanical manipulation. The electrical contact was made by a Pt/Ir metal-coated AFM tip that was

cold-welded to the metal top by applying a normal force of 50 nN along with an electrical current pulse of 1 mA for a duration of 1 s. The strong mechanical contact formed allowed us to apply lateral shear forces of up to ~200 nN and to induce a shear glide along a single basal plane within the graphitic stack[24] (see inset of Fig. 1b). The total shear force, $F_{total}$, is composed of a reversible restoring displacement force due to adhesion, $F_{adhesion}$, and to a smaller irreversible friction force, $F_{friction}$, that leads to the appearance of a force hysteresis loop[24]. The small magnitude of the measured $F_{friction}$ and the small force fluctuations observed (<10 nN) indicate that the sliding was done under superlubric conditions[24,28–30] (see Supplementary Note 1) and that the graphitic interface was twisted by a rotational mismatch of ~10 ± 5° along the individual slip plane within the stack[24,27]. Along with the mechanical actuation, we applied a DC bias voltage to the AFM tip and the current passing through the whole structure was measured using a pre-amplifier that collected the current from the HOPG substrate.

To validate the mechanical integrity of the tip-mesa contact, the stability of the interface under lateral manipulation, and the superlubric nature of the sliding, we first sheared the junction repeatedly from left to right by a distance that equaled the radius of the circular contact, while keeping the applied normal force exerted by the tip below 5 nN. The actual current measurement was then performed by shifting the upper mesa section starting from the fully overlapped position up to the complete removal of the top part (Fig. 1a).

During the shear process of the circular mesa the overall contact area reduces with the sliding distance, $x$, as follows (Fig. 1c):

$$S^{Bulk}(x) = 2\left(r^2 \cdot \cos^{-1}\left(\frac{x/2}{r}\right) - \frac{x}{2}\sqrt{r^2 - \left(\frac{x}{2}\right)^2}\right), \quad (1)$$

where $r$ is the mesa radius. As may be expected, this results in a gradual current reduction with the sliding distance (Fig. 1a) that could, in principle, be modeled by a simple electronic circuit consisting of four serial resistors (see Supplementary Note 2): (1) the interfacial resistance $R_{int}^{Bulk}$ that is assumed to scale inversely with the contact area, $S^{Bulk}(x)$; (2), (3) the constant ohmic resistance of the upper and lower graphitic mesa sections ($R_{Gr}$); and (4) the constant ohmic resistance of the measuring apparatus ($R_{Sys}$) including the tip-sample contact resistance, the AFM internal resistance, the spreading resistance of the graphite pillar, cables resistance, etc.[25]. Notably, the best fit that such a model can provide for the dependence of the measured current on the sliding distance cannot reproduce the experimental curve (full blue line in Fig. 1b). This indicates that apart from the bulk conductance contribution there should be an additional transport channel that scales differently with respect to the sliding distance.

Geometrically, for the circular interface considered, the only surface overlap contribution that does not scale as Eq. 1 with the sliding distance is the circumference associated with the junction edges. This suggests that the missing ingredient in the circuit discussed above would be the conductance contribution of the edge region. To check this hypothesis, we extend the serial circuit model discussed above by adding a resistor $R_{int}^{Edge}$ in parallel with $R_{int}^{Bulk}$ (Fig. 1D). $R_{int}^{Edge}$ would then scale inversely with the edge contact length (Fig. 1c):

$$L^{Edge}(x) = 4\left(r \cdot \cos^{-1}\left(\frac{x/2}{r}\right)\right). \quad (2)$$

Strikingly, the addition of the edge contribution results in excellent fit with the experimentally measured currents (full red line in Fig. 1b). Furthermore, a very good fit is obtained (especially at large sliding distances) even when we completely

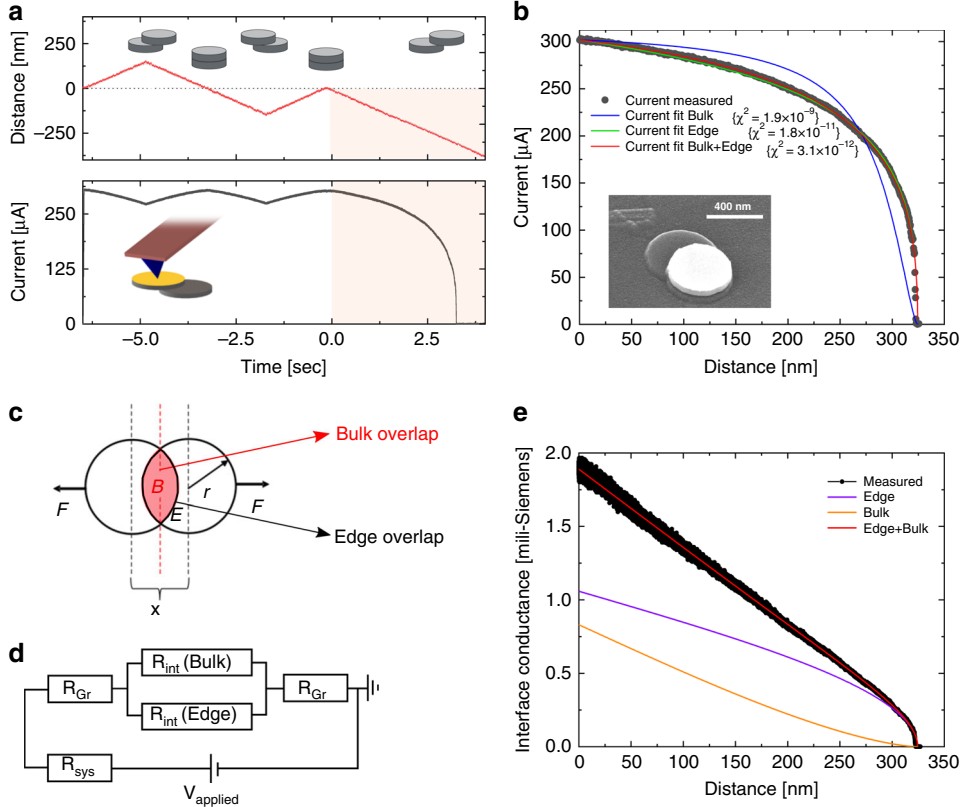

**Fig. 1 Experimental determination of edge and bulk interlayer transport contributions in a graphitic contact. a** Time dependence of the lateral displacement (top panel) and measured current (bottom panel) during interface shearing. The circular discs in the inset of the top panel describe the relative positions of the top and bottom graphite contacts during the shearing experiment. An illustration of the experimental setup is shown in the inset of the bottom panel. Up to $t = 0$ s, the top mesa is sheared partially by a distance equal to the mesa radius, whereas from $t > 0$ s, the mesa is sheared at a constant velocity of 100 nm s$^{-1}$ to a distance larger than the mesa diameter, resulting in a complete removal of the top mesa. **b** Measured current (full black circles) vs. lateral distance (data taken from (**a**) for $t > 0$) obtained at a bias voltage of 1V and fitted against an equivalent electrical circuit considering bulk (blue), edge (green) and bulk + edge (red) interlayer transport channels. The inset shows an SEM image of a partially sheared graphitic mesa over a single glide plane. **c** Schematic top view illustration of the sheared circular interface demonstrating overlap of both bulk and edge regions. **d** Equivalent electrical circuit considering conduction through both the bulk and edge regions in parallel resulting in the best fit to the measured current i.e. $\chi^2 = 3.1 \times 10^{-12}$. **e** Measured (black) and fitted (red) interface conductivities extracted by the model considering both edge and bulk contributions (circuit shown in **d**) that corresponds to the red curve in **b**. The total calculated conductivity (red) is the sum of the bulk (orange) and edge (purple) interface conductivities.

ignore the bulk contribution (full green line in Fig. 1b). This suggests that edge transport dominates the vertical conductance of the mesoscale interface.

In fact, the simple parallel circuit model suggested above allows us to distinguish between the bulk and edge transport contributions thus allowing for the quantitative evaluation of their relative importance. In Fig. 1e, we plot the bulk (full orange line) and edge (full purple line) conductance profiles ($G_{int}^{Bulk} = \left(R_{int}^{Bulk}\right)^{-1}$ and $G_{int}^{Edge} = \left(R_{int}^{Edge}\right)^{-1}$, respectively) as a function of the interface sliding distance for an applied bias voltage of 1 V. The sum of the two components (full red line) gives excellent agreement with the total interface conductance of the measured contact (full black line). The corresponding bulk and edge resistivities of $\rho_{Bulk} = 1.66 \times 10^{-10}$ Ω m$^2$ and $\rho_{Edge} = 3.54 \times 10^{-12}$ Ω m$^2$, respectively, (assuming an effective edge width of 2 nm as discussed below) demonstrate that edge conductance across the twisted graphitic interface is two orders of magnitude higher than its bulk counterpart.

We note that obtaining a full conductance-versus-shift profile for a given applied bias voltage, as presented in Fig. 1e, results in a breakdown of the measured mesa due to the eventual complete

removal of the top sheared stack from its bottom counterpart. Therefore, in order to study the conductance over a range of applied bias voltages, using the same contact, we performed the experiment by shearing the interface in steps of 5 nm and measuring a full current–voltage profile for each shift position (see Supplementary Note 3). Figure 2a presents 57 such current–voltage profiles, obtained using this procedure, for a bias voltage range of ±1 V spread according to their relative sliding positions. Slicing the current–voltage profiles along the distance axis give the desired current–distance curves at any given bias voltage value (Fig. 2b), for which we can perform the fitting procedure described above to obtain the edge and bulk current contributions (Fig. 2c). Finally, having at hand the separate contributions for any sliding distance and bias voltage allows us to extract the individual current–voltage profiles of the edge and bulk sections at a given shift position (Fig. 2d).

Focusing our attention on two extreme profiles, corresponding to a small lateral shift of 5 nm (full circles in Fig. 2d) and a nearly complete removal shift of 245 nm (empty circles in Fig. 2d), we make the following observations: (1) the overall current values of the small shift case are larger than those of its large shift counterpart mainly due to the higher overlap area; (2) in both cases the bulk current exhibits a transport gap of ~0.3 V

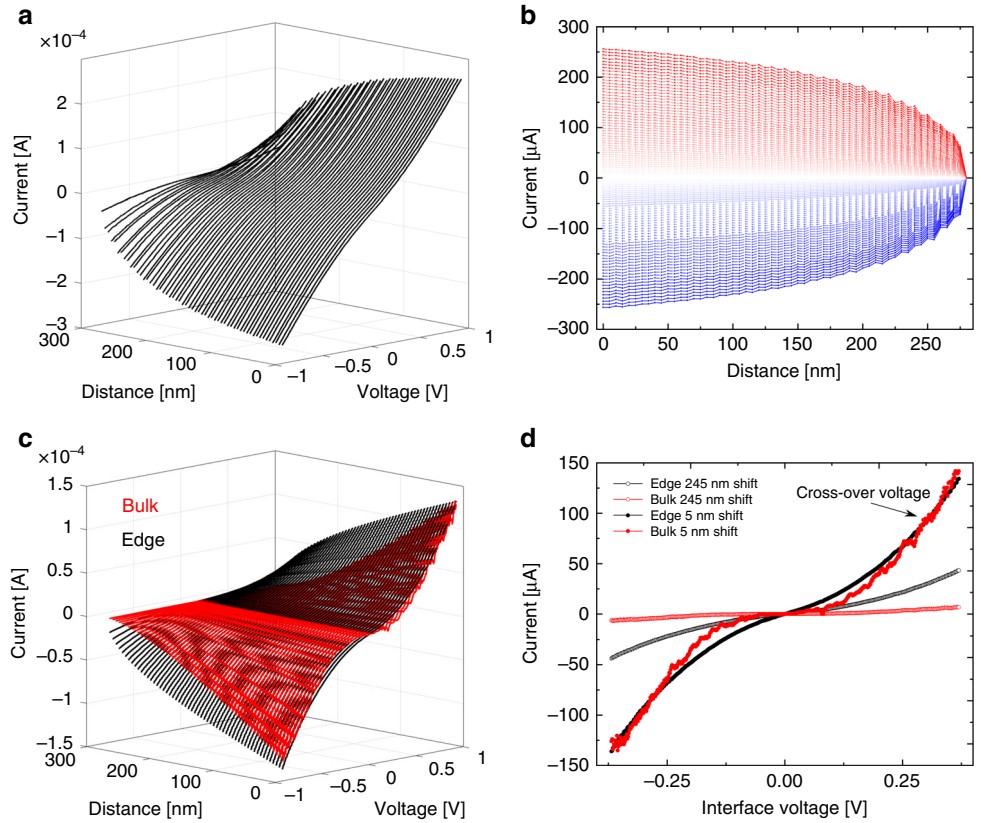

**Fig. 2 Reconstruction of the edge and bulk interface conduction vs. applied bias voltage. a** 3D diagram of the current vs. voltage profiles obtained at different relative sliding positions. **b** Reconstruction of the current vs. sliding distance for different applied voltages extracted from the data in **a**. Color code stands for negative to positive bias potential for dark blue to dark red, respectively. **c** Separated bulk (red) and edge (black) current vs. voltage profiles for different relative sliding positions. **d** Bulk (red) and edge (black) current vs. interface voltage characteristics extracted from **c** for sliding distances of 5 nm (full circles) and 245 nm (empty circles). The *i–V* profiles for sliding distances of 5 nm (full circles) show a cross-over from edge to bulk dominant transport for interface potentials exceeding 0.3 ± 0.05 V.

(attributed to the low measured current within bias range of ±0.15 V), whereas the edge current shows an Ohmic-like behavior (expressed by the linear *i–V* profile) mainly due to the large density of conducting edge states near the Fermi energy; and (3) while at smaller shifts the bulk and edge current contributions are comparable, with decreasing overlap area the latter becomes the dominant contribution, due to the different geometric dependence of the bulk (Eq. 1) and edge (Eq. 2) regions overlap functions (see, e.g., green and blue lines in Fig. 1e); (4) The *i–V* profiles corresponding to the small lateral shift of 5 nm (full circles) show a cross-over from edge to bulk dominant transport for interface potentials exceeding 0.3 ± 0.05 V (Fig. 2d).

**Theoretical analysis**. The significant role that edge transport plays in the overall transport characteristics of our junctions suggests that the prominent zero-energy edge states, typically localized at zigzag graphene terminations, are strongly involved in the transport process. To support this hypothesis, we performed interlayer transport calculations for a set of bilayer graphene flakes of dimensions 10–20 nm in diameter. The transport calculations were performed using the Landauer scattering formalism in conjunction with the non-equilibrium Green's function theory[31]. The electronic structure of the system was described by a tight-binding Hamiltonian that includes an exponentially decaying interlayer hopping integral[32]. More details regarding the calculations are provided in Supplementary Note 4. To demonstrate the importance of zigzag edge state for the transport properties of the system, we first consider hexagonal flakes,

terminated by either only zigzag or only armchair edges, stacked without any lateral shift at a misfit angle of 15° relative to the Bernal stacking, similar to the experimental value. While the highest occupied molecular orbitals (HOMO) of the hexagonal armchair flake (inset of Fig. 3b) uniformly distribute over the entire flake surface, the zigzag flake (inset of Fig. 3a) exhibits pronounced edge states, strongly localize at its circumference. Therefore, we define the edge region to be of width 0.4 nm at the circumference of the flake to include the majority of the electron density associated with these zigzag edge state. The total transmittance probability through the interface can then be split into edge and bulk region contributions (see Supplementary Note 4 for further details). In the main panels of Fig. 3a, b, we present the total transmittance probability (black) and its bulk-to-bulk (red), edge-to-edge (green) and bulk-to-edge plus edge-to-bulk (blue) components. Here, the edge-to-edge component, for example, is the probability that an electron impinging upon the edge of the lower flake will exit the upper flake from its edge region. A similar notion holds for all the other components. We observe three main differences in the transport characteristics of the zigzag and armchair bilayers: (1) the transmittance probability of the zigzag flake bilayer peaks at zero energy, where the density of zigzag edge states is maximal. On the contrary, the transmittance probability of the armchair flake bilayer is minimal at zero energy and grows away from this point; (2) the edge-to-edge component dominates the zigzag junction's transmittance probability, whereas the armchair junction's transmittance is mainly of bulk character, as may be expected from the uniform structure of its HOMO orbital; and (3) overall the low-energy transmittance of

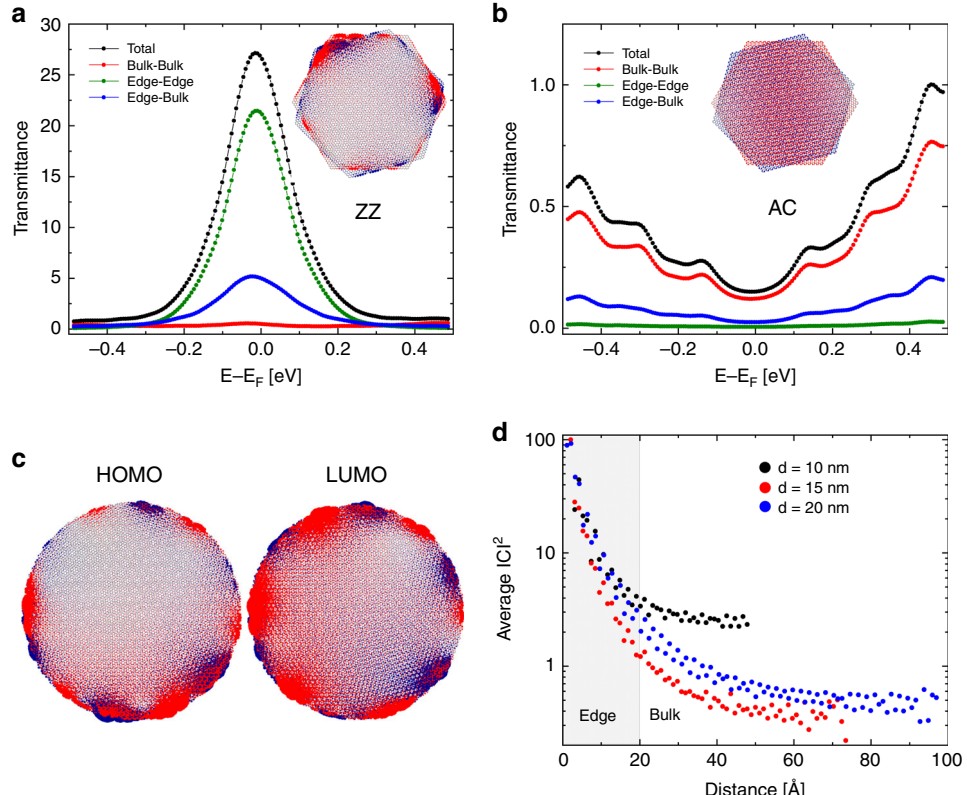

**Fig. 3 Calculated interfacial transmittance and wave-function distribution.** Calculated interfacial transmittance probability as a function of energy for **a** zigzag and **b** armchair hexagonal bilayer graphene junctions of side lengths of 7.6 and 7.8 nm, respectively, and angular mismatch of 15°. The total transmittance (black) is divided to show the individual contributions of bulk-bulk (red), edge-edge (green), and edge-bulk (blue). For the hexagonal flake calculations, edges are considered as 0.4 nm wide (see main text). Inset images show the highest occupied molecular orbitals (HOMO) of the **a** zigzag and **b** armchair hexagonal twisted bilayer interfaces. While for the armchair case, the wave-function uniformly distributes over the entire flake surface, the zigzag flake exhibits pronounced edge states, that are strongly localized at its circumference. **c** Calculated HOMO and LUMO orbitals for circular bilayer graphene structures with 15° rotational mismatch and a diameter of 20 nm (images for additional molecular orbitals next to the Fermi energy can be found in Supplementary Note 7). **d** Angularly averaged wave-function weights averaged over eight molecular orbitals in an energy range of ±0.44 meV around the Fermi energy, as function of radial distance for 3 different circular graphene bilayer interfaces of diameters 10, 15 and 20 nm and a twist angle of 15°. The edge region definition (containing ~99% of the edge state probability) in the 20 nm circular flake is depicted in gray.

the zigzag junction is considerably higher than that of its armchair counterpart.

Having demonstrated the important contribution of zigzag edge states to the transport characteristics of the system using hexagonal flakes, we now turn to examine experimentally relevant circular cross sections that consist of a disordered mixture of zigzag and armchair terminations. Figure 3c shows the weights of both the HOMO and LUMO of a 20 nm diameter bilayer circular junction on the different atomic sites, showing strong localization along the zigzag edge regions of the top (red) and bottom (blue) flakes. Despite this localization the edge states are not fully limited to the circumference of the flake but rather exponentially decay toward its bulk region. Therefore, to ensure that our flake models are sufficiently large to allow for substantial decay of the edge states toward the bulk we plot in Fig. 3d the size dependence of the angularly averaged molecular orbital weights (absolute squared expansion coefficients) as function of the distance from the flakes edges averaged over eight molecular orbitals in an energy range of ± 4.4 meV around the Fermi energy. The angularly averaged weights of the individual molecular orbitals are given in Supplementary Note 7. From Fig. 3d, it is apparent that the edge state decay function approaches convergence at flake diameters exceeding 15 nm. Therefore, we adopt a 20 nm diameter flake for all calculations described below. The edge region is defined to be 2 nm wide. This value is somewhat larger

than the value used for the hexagonal junctions discussed above due to the larger dimensions of the circular junction that allows us to effectively include up to ~99% of the edge states electron density (see gray marking in Fig. 3d). In order to extrapolate our calculations results for a 20 nm diameter circular cross-section junction to the experimental 300 nm diameter interface, we assume that the calculated transmittance and currents can be scaled according to the relative edge areas of width 2 nm such that $\frac{T_{300\,nm}^{Edge}}{T_{20\,nm}^{Edge}} = \frac{I_{300\,nm}^{Edge}}{I_{20\,nm}^{Edge}} = \frac{\pi \cdot [150^2 - 148^2]}{\pi \cdot [10^2 - 8^2]}$. A similar scaling procedure is applied for the calculated bulk transmittance (and currents) results, reflecting the increase in number of transmittance channels (and hence current) with increasing contact area.

This allows us to investigate the dependence of the transport mechanism on the interfacial misfit angle. Figure 4a presents the scaled bulk (red lines) and edge (black lines) transmittance probabilities for the Bernal (empty circles) and 15° rotated (full circles) circular interfaces. For clarity, we normalize each transmittance component by the corresponding total transmittance of the same system at any given energy point. For the Bernal stacked interface a clear dominance of the bulk transport over the edge contribution is evident throughout the energy range considered with some increase in edge transport near zero energy.

On the contrary, the 15° rotated system exhibits strong preference toward edge transport at the low-energy regime

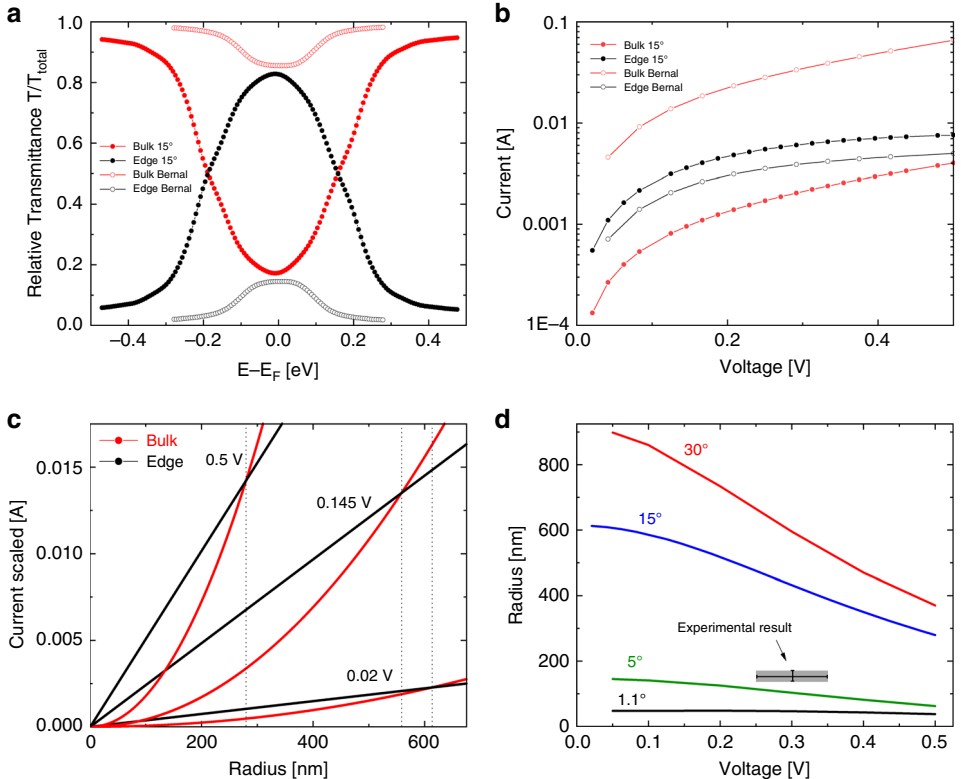

**Fig. 4 Calculated interfacial transmittance, current, and edge to bulk cross-over conditions. a** Calculated relative transmittance probability $T(E)/T_{total}(E)$ for Bernal (empty circles) and 15° twisted (full circles) 300 nm diameter circular graphene bilayer junctions, showing separately edge ($T_{Edge}(E)/T_{total}(E)$, black) and bulk ($T_{Bulk}(E)/T_{total}(E)$, red) contributions. **b** Calculated interlayer currents as a function of applied interlayer voltage for Bernal (empty circles) and 15° Twisted (full circles) graphene configurations, showing separately edge (black, including edge-bulk contributions) and bulk (red) contributions. Both the calculated transmittance curves and the currents are scaled to mimic a 300 nm diameter circular structure with 2 nm wide edges (see main text for further details). **c** Calculated edge (black) and bulk (red) currents as a function of circular bilayer graphene contact radius for different interlayer bias voltage drops. The different scaling laws for edge and bulk effective areas, result in a cross-over radius in which bulk conduction takes over edge conduction (marked by the dashed vertical lines). **d** Cross-over radius as a function of interlayer bias voltage (edge width is considered as 2 nm) for different interlayer angular configurations. Gray area marks the regime corresponding to our experimental study, i.e., contact diameter of 300 ± 10 nm and edge-to-bulk transport cross-over bias of 0.3 ± 0.05 V (see data for the 5 nm shifted junction in Fig. 2d).

(±0.2 eV), in good agreement with our experimental results presented in Fig. 2d, whereas at higher energies bulk transmittance takes over. Integrating over the transmittance probability within the Fermi transport window to obtain the currents we observe a similar behavior (Fig. 4b), where for the Bernal stacked interface (empty circles) the interlayer current is dominated by bulk (red) transport throughout the voltage range considered, whereas the 15° rotated system (full circles) edge current (black) is consistently higher than the bulk contribution in this bias range. We attribute this behavior to the fact that bulk transport depends strongly on the interlayer registry[23], whereas edge transport originates from the existence of pronounced edge states that depend more weakly on the twist angle. These transport characteristics are in good agreement with the predicted high density of zero energy zigzag graphene edge states and the experimentally observed interlayer conduction reduction in twisted bilayer graphene systems[23,33–36].

Clearly, since the edge states are limited to a narrow region in the vicinity of the contact circumference, upon increasing the surface area, the relative edge contribution to the overall transport diminishes and bulk transport should dominate for either Bernal or rotated junctions. Hence, we expect to observe a cross-over radius, $r_c$, that depends on the applied potential and misfit angle, in which the bulk current takes over its edge counterpart. To estimate this value, we repeated the edge and bulk current scaling procedure described above for increasing contact radii at several

bias voltages. Figure 4c shows the calculated edge (black) and bulk (red) current profiles as a function of the contact radius for three representative bias voltages, from which we can extract the cross-over value between edge and bulk current dominance. Figure 4d shows the cross-over radius as a function of applied voltage for the 15° rotated case of Fig. 4c (blue) as well as other misfit angles. Notably, despite the narrow width that the edge states span (~2 nm), edge transport dominance is expected to appear up to contact diameters of the order of 2 μm at the low bias–high misfit angle regime. The dark gray region in Fig. 4d marks the radius and edge-to-bulk transport cross-over bias ranges relevant to the present experiment (Fig. 2d). From this we can estimate the contact misfit angle between the upper and lower mesa sections in our experiment to be larger than 5° (green line) and lower than 15° (blue line). This is consistent with the value of ~10°, previously estimated from the analysis of friction measurements in similar interfaces[24].

## Discussion

Finally, we note that the edges of the various layers within the graphitic stack are most probably terminated by a variety of chemical terminations, in particular by different edge-oxidation schemes as a result of the oxygen based etching process. This issue has been previously studied, demonstrating that zigzag edge states survive various edge-oxidation schemes and their effect may even be enhanced by edge polarization[19]. Therefore, we do

not expect that edge chemistry (and especially edge-oxidation) will influence the qualitative nature of our general conclusions regarding interlayer edge transport.

The results presented above thus demonstrate how a careful analysis of the scaling laws of interlayer transport with respect to contact surface area can unveil the intricate interplay between edge and bulk interlayer transport contributions. Our finding that the former can dominate up to substantial incommensurate-contact dimensions suggests that, when performing interlayer transport experiments in graphitic interfaces and when designing graphene-based electronic devices, attention should be given to the role of edge states, which can govern the device performance.

## Methods

**Lateral force measurements.** The lateral shear force was measured using a tip velocity of 100 nm s$^{-1}$ during the sliding process in order to verify that sliding is performed under superlubric conditions, thus ensuring the existence of an angular mismatch at the bilayer graphene interface[24]. The shear force was evaluated using the relation $F = 2\sigma r$, applicable for small shear distances, where $r$ is the mesa radius and $\sigma = 0.227$ J m$^{-2}$ is the adhesion energy of graphite[24]. Results for the lateral force measurements are presented in Supplementary Note 1.

**Fitting procedure.** A numerical fitting procedure is employed in order to obtain the current vs. distance profile $I(x)$ and to extract the equivalent resistance of the sheared interface, $R_{int}$, composed of both edge ($E$) and bulk ($B$) transport contributions. $I(x)$ is calculated based on the equivalent electrical circuit depicted in Fig. 1d i.e. $I(x) = \frac{V_{applied}}{\{2 \times R_{Gr} + R_{sys} + R_{int}\}}$, where $R_{int} = \left[ \left(R_{int}^{Bulk}\right)^{-1} + \left(R_{int}^{Edge}\right)^{-1} \right]^{-1}$. The bulk and edge interfacial resistances are related to the lateral sliding distance, $x$, via $R_{int}^{Bulk} = \frac{\rho^{Bulk}}{S^{Bulk}(x)}$ and $R_{int}^{Edge} = \frac{\rho^{Edge}}{L^{Edge}(x)}$, where $S^{Bulk}(x)$ and $L^{Egde}(x)$ are given by Eqs. (1) and (2), respectively, and the corresponding resistivities $\rho_{Bulk}$ and $\rho_{Edge}$ serve as fitting parameters. The values of $R_{Gr}$ and $R_{sys}$ (which are considered constant throughout the sliding) are obtained from $I(x=0)$[25]. The interface voltage (in Fig. 2d) is extracted via: $V_{int} = I \cdot R_{int}$.

**Electronic transport calculations.** To evaluate the interlayer transport behavior of the graphitic interface three different model junctions were constructed: (1) armchair hexagonal bilayer junctions of side lengths of 7.8 nm; (2) zigzag hexagonal bilayer junctions of side lengths of 7.6 nm; and (3) circular bilayer junctions of 20 nm in diameter. The electronic structure of the various junctions was treated within the tight-binding approximation adopting the Hamiltonian parameterization of refs. [23,32] with no edge passivation. Such tight-binding calculations are known to be very successful in capturing electronic structure and optical characteristics of single and few-layered graphene[3,37–42] as well as their electronic transport behavior[43–45]. Furthermore, they capture the main physical characteristics of zigzag edge states in these systems[11,46–49]. Specifically, the tight-binding Hamiltonian adopted in the present work[32] was carefully calibrated against ab-initio calculations of twisted bilayer graphene[50] and was successful in rationalizing fine experimental findings on these systems[23]. This substantiates the suitability of the chosen tight-binding model to provide a good qualitative description of interlayer transport in twisted graphene interfaces in general and of the relative importance of edge and bulk states in the cross-layer transport in these systems, in particular. Details regarding the Landauer-based transport calculations, the separation of the transmittance probability into edge and bulk contributions, and convergence analysis of the results with respect to the model dimensions and its various parameters are given in Supplementary Notes 4–6.

## Data availability

The data that support the findings of this study are available from the corresponding author on request.

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

## Acknowledgements

E.K. gratefully acknowledge the Israel Science Foundation (ISF) for financial assistance and the RBNI for the nanofabrication facilities. E.K. thanks the Taub fellowship for leadership in science and technology, supported by the Taub Foundation and the Alon fellowship. O.H. is grateful for the generous financial support of the Israel Science Foundation under grant number 1740/13, The Ministry of Science and Technology of Israel under project number 3-16244, and the Center for Nanoscience and Nano-technology of Tel-Aviv University. A.O. gratefully acknowledges the support of the Adams Fellowship Program of the Israel Academy of Sciences and Humanities. We thank Urs Duerig, Armin Knoll and Abraham Nitzan for fruitful and stimulating discussions.

## Author contributions

E.K. conceived the experimental concept. E.K. and D.D. performed the experimental work. O.H. and A.O. developed and implemented the interlayer transport code. A.O. performed the numerical calculations of the interlayer transport. All authors participated in the data analysis and in the writing of the manuscript.

## Competing interests

The authors declare no competing interests.
