## [Peer Review File · Nature Communications]

REVIEWER COMMENTS

Reviewer #1 (Remarks to the Author):

This manuscript describing work on the edge and bulk transport in strained nanoscale vdW heterostructures is timely and relevant to the field. In general these are challenging nanoscale systems to probe experimentally, and this manuscript describes some evidence that edge transport can dominate over bulk interlayer transport in certain strained regimes. While the work seems interesting, I have the following concerns with the clarity and the underlying physical explanation and modeling of the results:

It's not clear why Fig. 1B shows very little difference between Edge and Bulk+Edge while in Fig. 1E the Edge and Bulk+Edge shows a large discernable difference.

A better schematic of the experimental setup and a complete (side-view) image of the structure corresponding to the text at the top of page 3 is needed. It's difficult for the reader to have a clear picture of what is going on.

I'm having a hard time understanding the 10 or 15 degree twist angle, and how it arises. From the etching technique as described in citation [24], shouldn't all the layers be AB stacked due to construction from a single bulk slab of HOPG? If this twist arises due to lateral straining, is it occurring over the entire thickness, or is there an individual slip plane within the stack (with the rest remaining unstrained)?

Do the authors know that covalent binding along the edges does not play a role? In the processing and shadow masking, is it possible for the edges to have a much greater conductivity due to possible covalent bonding?

Also, the model assumes the graphene layers shear smoothly. However, is it possible for the shearing to be accompanied by out of plane buckling and rippling? Such rippling could induce one-dimensional rippling (ie, point contact variations) that might mimic edge conduction modes.

The structures are 50 nm in height and the reported twist angle of about 15 degrees is presumably spread continuously through this thickness and the roughly 150 layers. Why then is the bi-layer modeling system relevant? Wouldn't 15 degrees over a bilayer be too large of an angular twist for the system?

Reviewer #2 (Remarks to the Author):

Authors report experimental and theoretical studies on mesoscale edge and bulk transports in twist graphitic interfaces.

Experimental data show detailed measurement on HOPG graphitic structure with 50 nm height and 300 nm diameter formed by reactive ion etching. Except this brief description of the experimental sample structure, there are no other data presented in the manuscript. There are no images of the sample shapes and chemical analysis of edge structures.

It was expected that the experimental samples are well represented by the model bilayer graphene structures with 10-20 nm diameter, without providing model validation by comparison with the

experimental samples.

The difference in size (50 nm vs. 0.7 nm bilayer height, 300 nm vs 10-20 nm diameter) and edge structures and chemical bonding (influenced by reactive ion etching) are not discussed at all, and the connection between the experimental data and modeling data is driven by conceptual association.

The separate discussion on experimental data and theory data look interesting, but it is not possible to gauge the meaningful connection between them. Specific details of the modeling data may or may not have any relevance to the experimental data. Without detailed and careful analysis of experimental samples, such connection cannot be established.

For example, 50 nm height represents ~ 147 graphene layers, authors implied the only one interface is sliding or twisting. It would be convincing to see the actual images of the specimen confirming such implications.

I recommend authors to provide detailed information about the experimental samples to establish any meaningful connection with the theoretical data. Of course, this would be a lot of work to do, but it is essential work to support claims in the manuscript. Due to this serious deficiency, the current manuscript is not publishable. If the authors successfully establish the experiment-theory connection supporting the claims in the manuscript, it would require a careful review to confirm the claimed connection between experimental data and modeling results.

Reviewer #3 (Remarks to the Author):

In this manuscript, the authors studied interlayer transport properties of bilayer graphene junctions. By precisely controlling of the displacement between two graphene layers, edge and bulk contributions to interlayer conduction are distinguished. Electronic structure calculations indicate that the edge contribution mainly comes from the edge state around the Fermi level at zigzag graphene edges. Results presented in this study is interesting and I recommend this manuscript to be published in Nature Communications after the following issues clarified.

1. In Figure 3d, the authors find that bulk current exhibits a transport gap while edge current shows an Ohmic-like behavior, why?
2. The authors suggest that the misfit angle in experiment is about 10 degree. Why not doing a TB simulation with this misfit angle and directly compare the simulated results with those obtained in experiment (Figure 3d)?
3. As a benchmark, is it possible to build a relatively small system and run a first principles NEGF simulation and compare the results with those obtained from tight binding simulation?

Referee 1:

1.1. *“This manuscript describing work on the edge and bulk transport in strained nanoscale vdW heterostructures is timely and relevant to the field. In general, these are challenging nanoscale systems to probe experimentally, and this manuscript describes some evidence that edge transport can dominate over bulk interlayer transport in certain strained regimes. While the work seems interesting, I have the following concerns with the clarity and the underlying physical explanation and modeling of the results:*

It’s not clear why Fig. 1B shows very little difference between Edge and Bulk+Edge while in Fig. 1E the Edge and Bulk+Edge shows a large discernable difference.”

Response to 1.1.: We thank the referee for his/her positive evaluation of our manuscript. We further thank him/her for pointing out this issue, which indeed may be a source of confusion. In Fig. 1B three fittings are provided: (i) The blue fitting relates to the equivalent circuit (Fig. 1D) considering only bulk transport, namely $R_{int}(Edge) = \infty$; (ii) The green fitting relates to the equivalent circuit (Fig. 1D) considering only edge transport, namely $R_{int}(Bulk) = \infty$; and (iii) the red fitting, where both $R_{int}(Edge)$ and $R_{int}(Bulk)$ are optimized. As we mention in the text, the latter that includes both bulk and edge contributions provides the best fitting with our experimental results, therefore it is used in the following analysis (shown in Fig. 1E). Namely, Fig. 1E considers only the last fitting (iii), where the red curve is consistent with the corresponding red curve of Fig. 1B. The purple and orange curves (green and blue in the previous version) here represent the components of the red fitting that relate to edge ($R_{int}^{-1}(Edge)$) and bulk ($R_{int}^{-1}(Bulk)$) transport, respectively, such that their sum gives the red curve.

It is also important to note that an additional reason for the small difference between the two profiles in Fig. 1B with respect to Fig. 1E is the fact that, while the former describes the current through the entire electrical circuit (including all resistors in series as describe in the main text and the schematic illustration in Fig. 1D), the latter describes the interface conductivity alone. Therefore, differences appearing in Fig. 1B appear to be smaller and the overall behavior is altered due to the influence of the additional components’ resistance.

To avoid confusion, we changed the colors of the edge and bulk component in Fig. 1E, and revised the figure caption that now reads: “(E) Measured (black) and fitted (red) *interface* conductivities extracted for the model considering both edge and bulk contributions (circuit shown in D) that corresponds to the red curve in Fig. 1B. The total calculated conductivity (red) is the sum of the bulk (orange) and edge (purple) interface conductivities”. Furthermore, we Italicized the word “interface” in this description. In addition, the supplementary information was revised accordingly.

1.2. *“A better schematic of the experimental setup and a complete (side-view) image of the structure corresponding to the text at the top of page 3 is needed. It’s difficult for the reader to have a clear picture of what is going on.”*

Response to 1.2.: We thanks the referee for this constructive comment. To address this issue and make our experimental setup description more transparent we have revised Fig. 1A by adding a schematic side view illustration of the experimental apparatus (see new inset in the revised manuscript).

1.3. *“I am having a hard time understanding the 10 or 15 degree twist angle, and how it arises. From the etching technique as described in citation [24], shouldn’t all the layers be AB stacked due to construction from a single bulk slab of HOPG? If this twist arises due to lateral straining, is it occurring over the entire thickness, or is there an individual slip plane within the stack (with the rest remaining unstrained)?”*

Response to 1.3.: We thank the referee for pointing out that this point requires better explanation. Indeed, the etched pillar comprises mostly of AB stacked layers. Nevertheless, the fact that we observe superlubric sliding indicates the stack includes at least one fault upon which the sliding occurs. This is supported by several previous studies on similar graphitic pillars including: “Observation of Microscale Superlubricity in Graphite”, Phys. Rev. Lett. 108, 205503; “Superlubricity of graphite”, Phys. Rev. Lett. 92, 126101 and “adhesion and friction in mesoscopic graphite contacts” *Science*, 348, 6235, 2015. In all these studies the sliding occurred along a single glide plane. Furthermore, based on the latter study (*Science*, 348, 6235, 2015) the force fluctuations level during the sliding process indicate that the twist angle in our experiments is within the range of $\sim 10^\circ \pm 5^\circ$.

To better convey this point we have modified the corresponding section in the main text that now reads “The small magnitude of the measured friction and the small force fluctuations observed (< 10 nN) indicate that the sliding was done under superlubric conditions^{24,28–30} (see Supplementary Information (SI) section 1) and that the graphitic interface was twisted by a rotational mismatch of $\sim 10^\circ \pm 5^\circ$ along the individual slip plane within the stack^{24,27}”. In addition, we have added an SEM image to the inset of Fig. 1B showing a single graphite pillar that is only partially sheared. The image shows that the sliding is done over a single glide interface located at approximately the middle of the graphite pillar. Finally, we have added an AFM (SI section 1) image of a fully sheared graphitic mesa showing that the exposed bottom mesa consists a single crystalline graphitic surface with an arithmetic average surface roughness, $R_a = 0.108$ nm.

1.4. “Do the authors know that covalent binding along the edges does not play a role? In the processing and shadow masking, is it possible for the edges to have a much greater conductivity due to possible covalent bonding?”

Response to 1.4.: This, indeed, is a very important point raised by the referee. We completely agree that the edges of the various layers within our stack are most probably terminated by a variety of chemical terminations. This results from the oxygen based reactive ion etching process that we utilized to construct the graphitic pillars. As the referee suggests, two effects of such terminations may influence our results. The first relates to the robustness of the conductive edge states towards different edge oxidation schemes. This issue has been previously studied demonstrating that zigzag edge states survive various edge-oxidation schemes and their effect may be even enhanced by edge polarization (see, e.g. *Nano Lett.* 2295, 7 (2007)). Therefore, we do not expect that edge chemistry (and especially edge oxidation) will influence the qualitative nature of our general conclusions regarding interlayer edge transport. The second issue regards the possibility of interlayer covalent bonding at the edges. We completely agree that edge-to-edge covalent bonding may influence the results at the fully eclipsed configuration. In contrast, once the interface shifts sideways, transport is dominated by edge-to-bulk conductance (as shown in the paper), where the probability of covalent bonding between dangling edge bond and the pristine graphitic surface reduces considerably. This is further supported by the superlubric behavior that our system exhibits that rules out massive interlayer covalent bonding.

To address this issue, we have added the following sentence to the main text: “Finally, we note that the edges of the various layers within the graphitic stack are most probably terminated by a variety of chemical terminations, in particular by different edge oxidation schemes as a result of the oxygen based etching process. This issue has been previously studied demonstrating that zigzag edge states survive various edge-oxidation schemes and their effect may even be enhanced by edge polarization¹⁹. Therefore, we do not expect that edge chemistry (and especially edge oxidation) will influence the qualitative nature of our general conclusions regarding interlayer edge transport.”

1.5. “Also, the model assumes the graphene layers shear smoothly. However, is it possible for the shearing to be accompanied by out of plane buckling and rippling? Such rippling could induce one-dimensional rippling (ie, point contact variations) that might mimic edge conduction modes.”

Response to 1.5.: This is a very interesting idea raised by the referee. Indeed, when shear forces are applied to bare graphitic surfaces, peeling and rippling effects can occur. Nevertheless, the situation in our experiments is quite different, as the sliding interface is buried deep inside the pillar and is supported by two thick graphitic slabs. This induces very strict constraints on the out-of-plane motion of the carbon atoms. Furthermore, our sliding interface is incommensurate thus the shear forces are expected to be very small

and hence rippling and buckling are highly unlikely to occur. One may think that some rippling may occur at the exposed surfaces during the sliding process. Nevertheless, we do not see any evidence for such rippling in the frictional behavior of the system, that demonstrates power law scaling of the friction forces with the contact area with an exponent of 0.3 as expected for incommensurate flat circular contacts (see e.g. “Scaling Laws of Structural Lubricity”, Phys. Rev. Lett. 111, 235502, 2013 and “adhesion and friction in mesoscopic graphite contacts” *Science*, 348, 6235, 2015). This is also supported by the fact that there is no experimental evidence of wear in our contacts. Furthermore, the bottom panel of Fig. 1A shows that the measured current is symmetric with respect to the fully-eclipsed configuration. Namely, for a given absolute shift value the current is the same when the top mesa shifts towards the center or away from it. If, in the former case, puckering (see e.g. “Frictional Characteristics of Atomically Thin Sheets”, *Science* 328, 76 (2010)) of the lower surface would occur in front of the sliding surface, the current would not be symmetric. Another experimental support for these claims comes from the fact that all current vs. shift distance scaling laws derived in this paper rely on the circular geometry of the interface. Random rippling and buckling effect would not obey such straight-forward scaling laws with the shift distance. Based on all the above, we believe that the rigid sliding assumption made in our simplistic model is valid.

To address this point, we have added an SEM image showing a partially sheared graphitic mesa demonstrating the bulky and robust nature of the graphitic interface (see inset in Fig. 1B). In addition, a discussion about the unlikelihood of ripples has been added to the revised SI in section 1.

1.6. “The structures are 50 nm in height and the reported twist angle of about 15 degrees is presumably spread continuously through this thickness and the roughly 150 layers. Why then is the bi-layer modeling system relevant? Wouldn’t 15 degrees over a bilayer be too large of an angular twist for the system?”

Response to 1.6.: This point is also related to point 1.3 above. As mentioned in our response there, the misfit angle of $10^\circ \pm 5^\circ$ does not span over the entire stack but rather occurs at a well-defined slip-plane. The rest of mesa, which is optimally stacked, serves as a mechanical support and as an electrical resistor connected in serial to the twisted contact. Therefore, a bilayer model is sufficient to capture the main physical features of the interlayer transport sliding-distance dependence. Furthermore, to demonstrate that the qualitative nature of our transport calculations is not strongly affected by this minimalistic bilayer model, we presented in the supporting information calculations for a quad-layer model system.

Referee 2:

2.1. *“Authors report experimental and theoretical studies on mesoscale edge and bulk transports in twist graphitic interfaces. Experimental data show detailed measurement on HOPG graphitic structure with 50 nm height and 300 nm diameter formed by reactive ion etching. Except this brief description of the experimental sample structure, there are no other data presented in the manuscript. There are no images of the sample shapes and chemical analysis of edge structures.”*

Response to 2.1.: We Thank the referee for pointing out that more visual description of the graphitic samples is needed to allow better understanding of the experimental setup. We therefore added an SEM image of partially sheared graphitic structures to the main text (Fig. 1A) and an AFM image to SI section 1 showing the exposed surface of a fully sheared mesa. The second concern regarding the edge chemical structure was also raised by referee 1. We addressed this point by adding the following section to the main text: **“Finally, we note that the edges of the various layers within the graphitic stack are most probably terminated by a variety of chemical terminations, in particular by different edge oxidation schemes as a result of the oxygen based etching process. This issue has been previously studied demonstrating that zigzag edge states survive various edge-oxidation schemes and their effect may even be enhanced by edge polarization¹⁹. Therefore, we do not expect that edge chemistry (and especially edge oxidation) will influence the qualitative nature of our general conclusions regarding interlayer edge transport.”**

2.2. *“It was expected that the experimental samples are well represented by the model bilayer graphene structures with 10-20 nm diameter, without providing model validation by comparison with the experimental samples. The difference in size (50 nm vs. 0.7 nm bilayer height, 300 nm vs 10-20 nm diameter) and edge structures and chemical bonding (influenced by reactive ion etching) are not discussed at all, and the connection between the experimental data and modeling data is driven by conceptual association.*

The separate discussion on experimental data and theory data look interesting, but it is not possible to gauge the meaningful connection between them. Specific details of the modeling data may or may not have any relevance to the experimental data. Without detailed and careful analysis of experimental samples, such connection cannot be established. For example, 50 nm height represents ~147 graphene layers, authors implied the only one interface is sliding or twisting. It would be convincing to see the actual images of the specimen confirming such implications.

I recommend authors to provide detailed information about the experimental samples to establish any meaningful connection with the theoretical data. Of course, this would be a lot of work to do, but it is essential work to support claims in the manuscript. Due to this serious deficiency, the current manuscript is not publishable. If the authors successfully establish the experiment-theory connection supporting the claims in the manuscript, it would require a careful review to confirm the claimed connection between experimental data and modeling

results.

Response to 2.2.: We thank the referee for raising this important point regarding the relation between the experiment and the computational model. In fact, this point was also raised by Referee 1. Regarding the thickness of the sample, we note that the dependence of the vertical transport of our mesas on the sliding distance is fully determined by a single slip-plane. To prove this, we followed the referees' suggestion and added an SEM image to the main text (Fig. 1A) showing that shearing in our system occurs at a single interface. The rest of mesa, which is optimally stacked, serves as a mechanical support and as an electrical resistor connected in serial to the twisted contact (see our recent publication *Nature Nanotech.* 11, 752–757 (2016)). Therefore, a bilayer model is sufficient to capture the main physical features of the interlayer transport sliding-distance dependence in our experiments. Furthermore, to demonstrate that the qualitative nature of our transport calculations is not strongly affected by this minimalistic bilayer mode, we presented in the supporting information calculations for a quad-layer model system.

As the referee mentions, the diameter of our model system is much smaller than that in the measurement. This, naturally, results from the computational burden associated with increasing the model system dimensions. Nonetheless, since we are interested in the relative contribution of edge states vs. bulk conduction, the main condition that the model system diameter should fulfil is that the zigzag edge states decay well into its bulk region. In such case, the qualitative nature of the relative importance of edge and bulk transport should be valid. To this end, we have performed detailed analyses of the decay of the edge states as function of system diameter (see Fig. 3d in the main text and SI section 7) showing that above 15 nm diameter the wave function decay into the bulk of the flake is well converged. Once such convergence was achieved, we could use simple scaling laws to compare our computational results to the experimental measurements, as done in Fig. 4.

To further support the validity of our model system for capturing fine details of the interlayer electronic transport characteristics of twisted interfaces, we note that in a recent publication (*Nature Nanotech.* 11, 752 (2016)) a very similar model system was used to successfully reproduce and rationalize coherent commensurate transport peaks appearing in the experiment. Therefore, we believe that our model system is well suited to capture the relevant qualitative transport behavior of the system.

Regarding the chemical nature of the edges and their structure. A similar point was raised also by Referee 1. In our experimental setup, the edges are cut into a circular structure using the oxygen plasma etching method. This implies that edge disorder and edge oxidation is present. To capture the disorder of the edges, our model system is also cut into a circular structure followed by the elimination of dangling bonds. The resulting structure has complex edge structure consisting of random zigzag and armchair sections, which mimics well the experimental situation on a smaller scale, as discussed above. Regarding the chemical nature of the edges, please see our above response to point 2.1 of referee 2 as well as our response to point 1.4 raised by referee 1.

Referee 3:

3.1. *“In this manuscript, the authors studied interlayer transport properties of bilayer graphene junctions. By precisely controlling of the displacement between two graphene layers, edge and bulk contributions to interlayer conduction are distinguished. Electronic structure calculations indicate that the edge contribution mainly comes from the edge state around the Fermi level at zigzag graphene edges. Results presented in this study is interesting and I recommend this manuscript to be published in Nature Communications after the following issues clarified.*

1. *In Figure 3d, the authors find that bulk current exhibits a transport gap while edge current shows an Ohmic-like behavior, why?”*

Response to 3.1.: We thank the referee for her/his positive response. Since Figure 3d discussed another aspect that is not related to the referee’s question, we assume that s/he is referring to Fig. 2D. As the referee states, the individual current vs. voltage curves for edge and bulk show distinctively different behavior. The current vs. voltage profile of the edge transport is much more linear with respect to the bulk profile. This indicates that the former exhibits a large density of transmitting edge states near the Fermi energy thus resulting in an ohmic behavior. In contrast, the latter exhibits a nonlinear profile, where in the bias range of ± 0.15 V a very low current is measured, whereas at larger voltages the current increases rapidly and eventually exceeds that of the edge contribution. This is indicative of the fact that near the Fermi energy the density of bulk states is considerably lower than that of the corresponding edge states. This is also consistent with the relative contribution of the bulk and edge regions to the overall transmittance probability as shown in Fig. 3A, B.

Following the referee’s question, we have added the following line to the main text: “in both cases the bulk current exhibits a transport gap of ~ 0.3 V (attributed to the low measured current within bias range of ± 0.15 V), whereas the edge current shows an Ohmic-like behavior (expressed by the linear i-V profile) mainly due to the large density of conducting edge-states near the Fermi energy;”.

3.2. *“The authors suggest that the misfit angle in experiment is about 10 degree. Why not doing a TB simulation with this misfit angle and directly compare the simulated results with those obtained in experiment (Figure 3d)?”*

Response to 3.2.: We would like to note that the experimental misfit angle is estimated to be $10^\circ \pm 5^\circ$. As the referee noticed, in some of the simulations we opted to use the upper bound of this range. This is done to include a larger number of moiré supercells within the bulk region to better mimic the experimental scenario. We stress that in Fig. 4D we present results obtained for a wide range of misfit angles ($1.1^\circ - 30^\circ$), to provide a complete picture of the edge vs. bulk transport characteristics.

3.3. “As a benchmark, is it possible to build a relatively small system and run a first principles NEGF simulation and compare the results with those obtained from tight binding simulation?”

Response to 3.3.:

We thank the referee for raising this point. As the referee mentions, in order to perform first-principle transport calculations using the non-equilibrium Greens function formalism one would have to construct a small model system. While this could provide information regarding the total cross-layer transport in the model system, it would not be able to address the main issue discussed in this paper, namely, the interplay between bulk and edge transport in graphitic interfaces. As mentioned in our response to point 2.2 raised by Referee 2, in order to be able to address this issue one has to construct a sufficiently large model system such that the edge states decay well into the bulk. This requires very large model systems (at least of the order of those used in our tight-binding calculations) that are heavily challenging for first-principles electronic transport calculations.

It should be noted, however, that tight-binding calculations are known to be very successful in capturing electronic structure and optical characteristics of single and few-layered graphene (see e.g. *AIP Advances* 7, 075212 (2017); *Rev. Mod. Phys.* 88, 025005 (2016); *Phys. Rev. B* 89, 165430 (2014); *Appl. Phys. Lett.* 102, 253506 (2013); *Rev. Mod. Phys.* 81, 109–62 (2009); *Eur. Phys. J. Special Topics* 148, 5–13 (2007); *Phys. Rev. B* 66, 035412 (2002) and many others) as well as their electronic transport behavior (see e.g. *Phys. Rev. B* 101, 245407 (2020); *J. Phys.: Condens. Matter* 30 364001 (2018); *J. Appl. Phys.* 113, 144506 (2013), and many others). Furthermore, they capture the main physical characteristics of zigzag edge states in these systems (see e.g. *Phys. Rev. Lett.* 84, 3390 (2000); *Phys. Rev. B* 62, R16349(R) (2000); *Phys. Rev. B* 59, 8271 (1999); *Phys. Rev. B* 59, 9858 (1999); *Phys. Rev. B* 54, 17954 (1996)).

We emphasize that the tight-binding Hamiltonian adopted in the present work (*Appl. Phys. Lett.* 103, 243114 (2013)) was specifically calibrated against ab-initio calculations of twisted bilayer graphene (*Phys. Rev. Lett.* 109, 236604 (2012)) and was shown to rationalize fine experimental findings on these systems (*Nature Nanotech.* 11, 752 (2016)).

All of the above thus substantiates the suitability of the chosen tight-binding model to provide a good qualitative description of interlayer transport in twisted graphene interfaces in general and of the relative importance of edge and bulk states in the cross-layer transport in these systems.

To clarify this point, we added the following sentences to the methods section of the main text:

“Such tight-binding calculations are known to be very successful in capturing electronic structure and optical characteristics of single and few-layered graphene^{3,38–43} as well as their electronic transport behavior^{44–46}. Furthermore, they capture the main physical characteristics of zigzag edge states in these systems^{11,47–50}. Specifically, the tight-binding Hamiltonian adopted in the present work³² was carefully calibrated against ab-initio

calculations of twisted bilayer graphene⁵¹ and was successful in rationalizing fine experimental findings on these systems³³. This substantiates the suitability of the chosen tight-binding model to provide a good qualitative description of interlayer transport in twisted graphene interfaces in general and of the relative importance of edge and bulk states in the cross-layer transport in these systems.”.

REVIEWERS' COMMENTS:

Reviewer #1 (Remarks to the Author):

It is fine.

Reviewer #2 (Remarks to the Author):

Authors have addressed the main comments raised in the previous review. The additional experimental and theoretical data help to address the concern on the conceptual gap between the experimental and modeling data. They are satisfactory and the revised manuscript would be publishable.

Reviewer #3 (Remarks to the Author):

The revised manuscript is recommended to be published in NC